# The Thermo-Phase Change Reactivity of Textile and Cardboard Fibres in Varied Concrete Composites

Robert Haigh , Malindu Sandanayake * , Paul Joseph , Malavika Arun , Ehsan Yaghoubi , Zora Vrcelj and Soorya Sasi

Institute for Sustainable Industries and Liveable Cities, Victoria University, Melbourne, VIC 3011, Australia; robert.haigh@vu.edu.au (R.H.); paul.joseph@vu.edu.au (P.J.); malavika.arun@vu.edu.au (M.A.); ehsan.yaghoubi@vu.edu.au (E.Y.); zora.vrcelj@vu.edu.au (Z.V.); soorya.sasi@vu.edu.au (S.S.)
* Correspondence: malindu.sandanayake@vu.edu.au; Tel.: +61-399-194459

**Abstract:** The building and construction industry heavily relies on the use of concrete and cementitious composites due to their exceptional attributes, including strength and durability. However, the extensive use of these materials has led to significant environmental challenges, including resource depletion, carbon emissions, and waste accumulation. In response to these challenges, recent advancements in fibre cementitious composites have shown promise in mitigating these detrimental effects. The integration of waste materials to supplement manufactured fibres represents a promising development in reinforced concrete composite materials. Waste materials like textiles and cardboard are emerging as potential fibre supplements in cementitious composites. While these materials have primarily been investigated for their mechanical characteristics, understanding their thermal properties when applied in construction materials is equally crucial. Incorporating fibres within composite designs often requires matrix modification to reduce degradation and enhance fibre longevity. This study aims to investigate the thermo-phase change properties of both textile and cardboard fibres within varied concrete matrices. Additive materials offer a range of advantages and challenges when used in composite materials, with additional complexities arising when incorporating fibre materials. Understanding the thermal reactivity of these materials is crucial for optimizing their application in construction. This study demonstrates the potential of waste fibres used with gypsum, metakaolin, and silica fume as matrix modifiers in concrete. This research provides valuable insights for future studies to explore specific material combinations and investigate complex fire testing methods, ultimately contributing to the development of sustainable construction materials.

**Keywords:** cardboard; concrete; composite; gypsum; kraft-fibres; metakaolin; silica-fume; textile-fibres; thermal

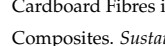


## 1. Introduction

The building and construction industry significantly requires the use of concrete and cementitious composites. This is due to the unparalleled attributes encompassing strength, durability, and versatility of the material [1]. Utilizing the enormous amount of these materials creates significant environmental damage including, resource depletion, carbon emission creation and waste accumulation [2]. Advancements in fibre cementitious composites shows a potential to reduce some of these detrimental effects. The adoption of waste materials to supplement manufactured fibres is an encouraging development in reinforced concrete composite materials. Currently, inorganic materials such as basalt fibres, steel fibres, rubber tyre fibres and hybrid cement-based composites have emerged as promising building materials [3–6]. The use of these fibres has reduced crack propagation and enhanced durability characteristics. Other waste materials such as plastics, textiles, and cardboard are emerging as possible fibre supplements in cementitious composites [7–9]. These materials have predominantly been investigated for their mechanical

characteristics; however, it is critical to understand their thermal properties when applied in construction materials. Moreover, it is important to have a thermal measurement of bespoke materials in varied matrices before the materials are applied for extensive testing. Thermal investigations of single fibre types in concrete have been studied; however, further investigations including the combination of multiple fibres in varied concrete matrices are rarely undertaken.

At the time of this study, few researchers have thermally investigated the use of multiple fibre types in one concrete composite mix design. Koří et al. [10] experimentally investigated the use of basalt and polypropylene fibres in reactive powder concrete (RPC). Their research demonstrated that applying high temperatures to the fibre concrete materials improved the spalling effect. This was shown when the polypropylene fibres deteriorated, increasing permeability. Moreover, this created water vapor escape paths that contributed to the internal pressure release capacity. Basalt fibres were shown to accommodate thermo-mechanical stress. Increasing the applied temperature to 20–1000 °C reduced the mechanical strength of the composites; however, the decline of the mechanical strength was linear, demonstrating slower degradation attributes rather than abrupt strength reduction. This further signifies that the synthetic fibres contributed to creating additional pathways for internal pressure to be released. Zhang et al. [11] investigated the use of both steel and polypropylene in concrete composites. Their research determined that thermal conductivity increases with the increasing geometry of polypropylene fibres, and the promoting effect caused by conductive steel fibres dominated the fibre concrete composites. The random dispersion of the polypropylene micro-fibres increased the transition pores and interfacial transition zones, thus demonstrating that the dispersion orientation and geometry of fibres can contribute to changes in thermal conductivity. It is important to note that partially supplementing cement can contribute to thermal phase changes and that fibre integration can be one thermal variable.

The use of additives is common practice to increase the strength, workability, durability, physical characteristics, and setting time of concrete [12]. However, in recent decades, alternative materials have also been used to increase sustainable measures via the integration of waste materials [13]. Namely, fly ash (FA) and ground blast furnace slag (GBFS) have been commonly adopted industrial waste materials [14]. These materials are also called supplementary cementitious materials (SCMs) because of their pozzolanic material properties [15]. Numerous studies show FA and GBFS geopolymer concretes have integrated sodium silicate and sodium hydroxide as a main activator [16–18]. Other SCMs used in composite designs are silica fume (SF) and metakaolin (MK) [19]. MK has been applied with fibres in previous studies [20–26]. It has been shown that MK modifies the matrix of the cementitious materials by reducing the particle size of the materials, exhibiting a denser matrix. When MK is used with fibrous materials, there is a stable interface on the fibre walls due to a reduced degradation occurring from the high alkalinity of cement. This can result with higher stress yields of fibre composites because the fibres have maintained their strength. Moreover, when the fibres maintain agility in the matrix, reduced crack propagation occurs, which can enhance the durability of the composite. However, limited research details the thermal reactivity of MK with fibres in a concrete matrix. Another issue with MK is that fibres may not necessarily anchor themselves within the matrix when MK is present, leaving the matrix more susceptible to fibre pull-out in these composite designs. Studies using MK and natural fibres detail similar findings [27–29]. This research indicates that MK can reduce chloride ion penetration, which can enhance the durability of the materials. The reduction in ion transferability can be due to the finer pore structure of MK.

Similarly, silica fume (SF) can also increase the mechanical function of fibre cementitious composites. Gencil et al. [30] demonstrate that a 15% SF integration can increase the mechanical strength of basalt fibre composites by 46%. Moreover, their findings showed that basalt fibre integration of up to 3% enhanced flexural strength by 88%. However, it was found that the integration of 15% SF made the composite material brittle, localizing macro-

cracks compared to the formation of microcracks. Kathirvel and Murali [4] discovered the use of 30% SF increased the compressive strength by 34.6%. This was shown by the SF particle size filling small voids within the matrix, lessening the amount of water required for hydration and thus leading to more compact development of the hydrating gels. Additionally, decreasing the porosity of the composite reduces the thermal conductivity. Lower thermal conductivity minimizes the speed of heat transfer, which can contribute to the improvement of insulating properties. It is important to note that the integration of high SF content can provide both positive and negative effects. This is dependent on the fibre type in composite designs. For example, Bagaria and Juneja [31] discovered that a maximum of 16% SF is mechanically viable with 0.8% polypropylene fibres. This is because the adhesion between the fibres and the matrix can vary, creating voids surrounding the interfacial zone of the fibre. If the fibres lose adhesion in the matrix, voids are created, reducing mechanical strength and increasing heat transfer. Therefore, it is critical to experimentally investigate multiple mix designs when using additive materials. SF reacts with calcium hydroxide $(Ca(OH)_2)$ in cement hydration, forming additional calcium silicate hydrate (C-S-H) gel. This reaction consumes less water and generates less heat during concrete curing. This can be critical in large concrete structures where controlling the temperature of the material is important.

Gypsum is another commonly used construction material due to its fire-resistant properties. However, the material is often neglected in cementitious composite designs. Several studies have highlighted key findings when integrating gypsum in composite materials [32–37]. Gypsum also contains small particles that can enhance a homogenous matrix. Having high efficiency of agglomeration in a matrix offers fibres the opportunity to de-bond gradually, rather than abruptly. This factor can increase stress yield but also maintain a stable composite material. The high thermal resistance of gypsum is its most prominent feature; this can also increase the setting time of the composite. Delaying curing can have additional mechanical effects which have not been outlined in research. Other research with natural fibres demonstrates that gypsum requires a higher water content compared to traditional additive materials [38–40]. This can then increase shrinkage and negatively affect the mechanical strength of the composite materials.

Additive materials pose an array of positive and negative affects when used in composite materials. However, the variabilities increase when using additional fibre materials. The synthetic chemical microstructure of textile materials can have varied reactions under applied heat. When comparing to cardboard fibres that are natural fibres, the reaction can vary greatly and ultimately create additional thermal consequences. Materials that maintain thermal control when exposed to high temperatures, will exhibit better fire performance. This study aims to demonstrate the use of the waste fibre materials in concrete and highlight their reactivity in varied matrices. Therefore, the following research will focus on using three additive materials: gypsum, MK, and SF. Moreover, this study will show the thermal reactivity when using the waste materials in concrete in conjunction with the suggested additives. The effect of the additive materials and how they react with the waste fibres will be experimentally investigated. Therefore, future research can focus on specific combinations derived from this study to investigate complex fire testing methods.

## 2. Methodology

Figure 1 illustrates the methodology constructed for this study. As shown, textile and cardboard (kraft fibres) are the fibres chosen to act as partial cement substitutions. MK, gypsum, and SF are the additive materials that supplement the altered percentages of cement. The mix design variations are further shown in Table 1. In total, there are 16 mix designs that are experimentally investigated for their thermal properties.

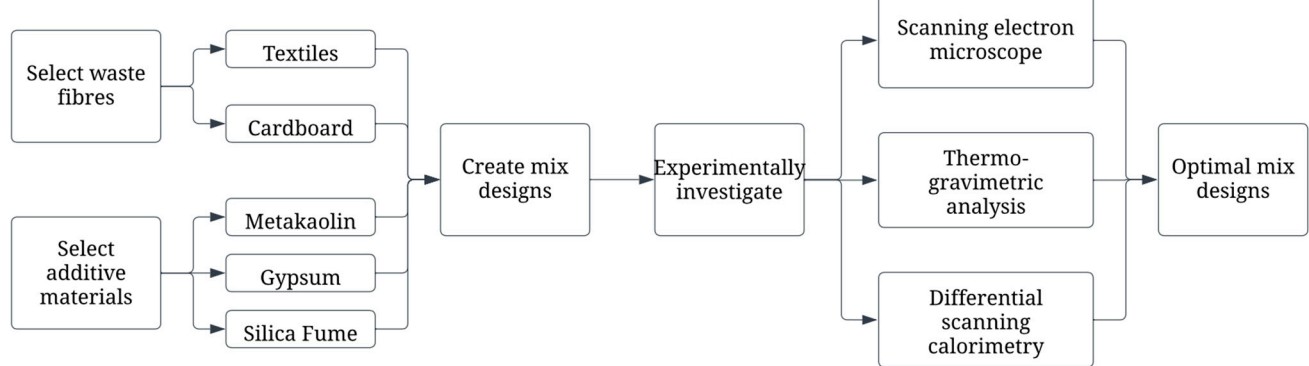

**Figure 1.** Methodology.

**Table 1.** Percentage of additive and fibrous materials.

| Mix Code | OPC (Weight) | Polyester Textile (Volume) | Silica Fume Kraft Fibres (Volume) | Gypsum (Weight) | SF (Weight) | MK (Weight) |
|---|---|---|---|---|---|---|
| S1 | 100 | | | | | |
| S2 | 90 | 5 | | 5 | | |
| S3 | 85 | | 5 | 5 | | |
| S4 | 95 | | | 5 | | |
| S5 | 95 | | | | | 5 |
| S6 | 95 | | | 2.5 | | 2.5 |
| S7 | 95 | | | | 5 | |
| S8 | 90 | | | | 5 | 5 |
| S9 | 92.5 | | 2.5 | | 2.5 | 2.5 |
| S10 | 90 | 5 | | | 5 | |
| S11 | 90 | 5 | | | | 5 |
| S12 | 85 | 5 | | | 5 | 5 |
| S13 | 85 | | 5 | | 5 | 5 |
| S14 | 87.5 | 2.5 | 5 | | | 5 |
| S15 | 87.5 | 2.5 | 5 | | 5 | |
| S16 | 85 | 5 | 5 | | 2.5 | 2.5 |

## 3. Materials

The main constituent materials utilized to reduce the proportion of OPC in the mixture formulations are waste corrugated cardboard, polyester textile fibres, MK, SF, and gypsum. Waste cardboard undergoes a transformation process into a fibrous substance called kraft fibres (KFs). KFs are lignocellulose, originating from plants and trees [41]. The incorporation of SF in the KF walls functions as a technique for modifying fibres. The procedure commences by breaking down the waste cardboard into a pulp through immersion in water and rotational mixing. Subsequently, a slurry of SF is applied to the KFs, yielding silica fume kraft fibres (SFKFs). The fibre then undergoes a process to eliminate moisture and are combined within a rotating mixer, creating a fibrous composition.

High vis construction vests are used as the textile material and polyester fibre. This material is a commercially available polyethylene terephthalate. The vests are shredded multiple times and turned into a fine fibrous material. The fibres are then added to the composites based on the sample weight percentage ratio. Figure 2 illustrates both textile and KF materials ready for concrete integration, and the SF employed conforms to the specifications of the Australian Standard AS/NZS 3582.3 [42] for silica fume used in cementitious materials. The MK, utilized as a partial replacement for cement, adheres to the criteria outlined in ASTM C-618 [43] Class N specification for natural and calcined pozzolans. The OPC used is in accordance with AS/NZS 3972 [44] and is included in the mix design as the primary element for pozzolanic reactivity. Coarse and fine aggregates, sourced locally, are incorporated into the specimens following the guidelines of AS/NZS 1141.6.2 [45] and AS/NZS 1141.5 [46], respectively. Table 2 details the chemical composition of the additive materials [47–50].

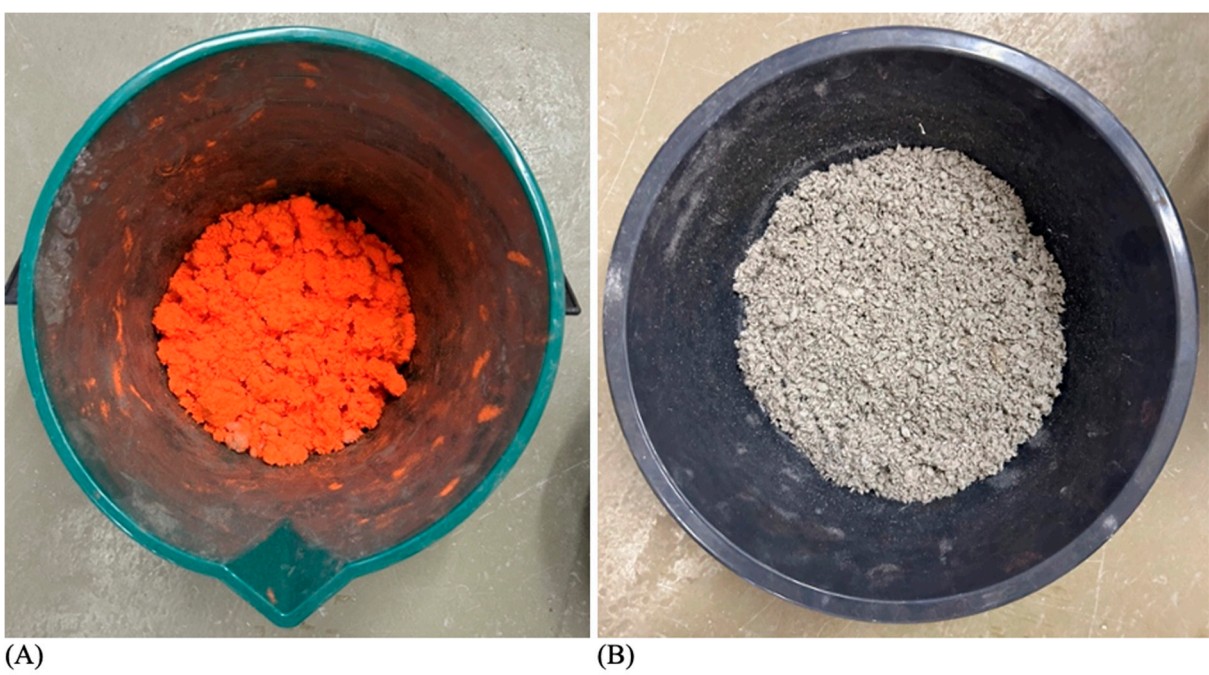

**Figure 2.** (**A**) Textile fibres and (**B**) kraft fibres.

**Table 2.** Chemical composition of pozzolanic materials.

| Chemical | Material Component% | | | |
|---|---|---|---|---|
| | OPC | Gypsum | SF | MK |
| $SiO_2$ | | | $\geq$0.3–<1 | |
| $Al_2Si_2O_5(OH)_4$ | | | | 97.5–100 |
| $CaSO_4$ | | 60–100 | | |
| CaO | >92 | | | |
| Silica, amorphous, fumed, cryst.-free | | <1 | $\geq$75–<100 | <0.1 |
| $CaSO_4.2H_2O$ | 3–8 | | | |
| $CaCO_3$ | 0–7.5 | | | |

Three 100 mm × 200 mm concrete cylinders are prepared for each mix design. The preparation of the samples involves the use of employing a mortar mixer and adhering to the directives of AS/NZS 1012.2 [51]. The materials are initially dry-mixed for five minutes to ensure proper agglomeration. Water is then introduced and mixed for an additional five minutes to complete the sample preparation process in accordance with AS 1379–2007 [52]. It is important to note that the fibres are slowly dispersed within the mortar-mixing during the mixing of all dry materials, including the additives. Once the mixture is ready, it is layered into moulds in a three-tier arrangement. After each layer, a steel rod is used to compress the material twenty times before the subsequent layer is added. The moulds are maintained at a room temperature of approximately 20 °C for 24 h before being transferred to curing baths. Slump tests are conducted per the protocols outlined in AS/NZS 1012.3.1 [53].

## 4. Testing Procedure

### 4.1. Scanning Electron Microscope (SEM)

To analyse the samples' microstructure, a combination of scanning electron microscope (SEM) and energy-dispersive X-ray spectroscopy (EDS) techniques were utilized. The Phenom XL G2 Desktop SEM from ATA Scientific Instruments, Caringbah, NSW, Australia, was employed for conducting the microstructure observations. Sample preparation involved the use of a diamond cutting saw to specimens with dimensions measuring 2–4 mm in height and 4–6 mm in diameter. It is important to note that the images obtained were from samples that were mechanically tested for their compressive strength. The

images highlight the reactivity of the fibres within the composite for their physical and microstructural characteristics.

### 4.2. Thermo-Gravimetric Analysis (TGA)

The TGA analyses of the specimens were conducted in a nitrogen environment, employing a heating rate of 10 °C/min. The temperature ranged from 30 °C to 900 °C. These analyses were carried out using a Mettler-Toledo instrument, with a gas flow rate of 50 mL/min. Samples sizes were approximately 9–10 mg each time, using a 70-µL silica crucible, with triplicate runs, optionally, to check the reproducibility, which was found to be ±0.00001 g. The selection of the 60 °C/min heating rate was motivated by the intention to enable direct comparisons with the outcomes obtained from pyrolysis combustion flow calorimetry (PCFC) investigations.

### 4.3. Differential Scanning Calorimetry (DSC)

To elucidate phase transitions and understand the energy dynamics inherent in pyrolytic processes, differential scanning calorimetry was conducted. The experimental runs were executed using a Mettler-Toledo instrument, DSC 1 STAR$^e$ system, as a stand-alone unit for the DSC tests. These analyses took place within a nitrogen atmosphere, employing a heating rate of 10 °C/min over the temperature range of 30 °C to 550 °C. The conditions of the analyses are as follows: sample sizes approximately 9–10 mg each; 50-µL aluminium crucible with a lid having a pin hole for venting the gaseous substances, if any, upon decomposition; triplicate runs, optionally, to check the reproducibility; ±0.00001 g; 10 K/min as the heating rate; gas flow: nitrogen gas at the rate of 20 mL/min. It is important to note that the TGA and DSC can be run simultaneously; for better evaluations, the Mettler Toledo was employed as a stand-alone unit for the DSC and TGA runs.

## 5. Results and Discussion

### 5.1. Microstructure Investigations

The microstructure of the fibrous samples was investigated via SEM. Figure 3 illustrates SEM images of the fibre samples. As shown, the textile fibres were approximately 15 µm in width, whereas the SFKFs were approximately 5–25 µm. The shape and size of SFKFs vary greatly due to being natural fibrous materials. Textile fibres are manufactured for uniformity. The SEM image settings for Figures 3 and 4 contained a magnitude (mag) of 430× factor weight (FW) 1.21 mm, high voltage (HV) 15 kV, detection (Det) BSD full, within design (WD) 13.342 mm and pressure (Pres) 0.10 Pa.

The SEM images shown in Figure 4 illustrate the effect of the varied concrete matrices on both textile and cardboard fibres. Figure 4A S13 corresponds to 5% SFKFs, 5% SF, and 5% MK. Figure 4B S10 demonstrates 5% textile fibres with 5% SF. Figure 4C S16 illustrates 5% textiles, 5% SFKFs, and 2.5% each for both SF and MK. Lastly, Figure 4D shows 5% textile fibres with 5% MK. As shown in Figure 4A, the SFKFs remained moderately undamaged. The outer fibre wall showed minor petrification, which is critical for the longevity of the fibre. This is mainly due to the pre-treatment of SF, which enhances service life of the fibres by mitigating the attack of calcium hydroxide ($Ca(OH)_2$) within the matrix. Moreover, SF reacts with $Ca(OH)_2$ to form additional calcium silicate hydrate (C-S-H) which can densify the concrete matrix and enhance strength characteristics. The integration of SF and MK as a partial cement replacement can enhance the durability of concrete materials, which in turn protects fibre integration. This is shown when SF and MK reduce the permeability of the composite and minimizing the creation of $Ca(OH)_2$ [54,55]. SF is effective in mitigating the risk of the alkali–silica reaction (ASR) which can deteriorate concrete materials [56,57]. SF reacts with the free alkalis in cement, effectively binding them together and reducing their availability for reaction with reactive silica in the aggregates. This reduction in free alkalis limits the potential for ASR to occur. Moreover, preventing ASR reduces the expansion and cracking of concrete, ultimately protecting the fibres in the matrix [58].

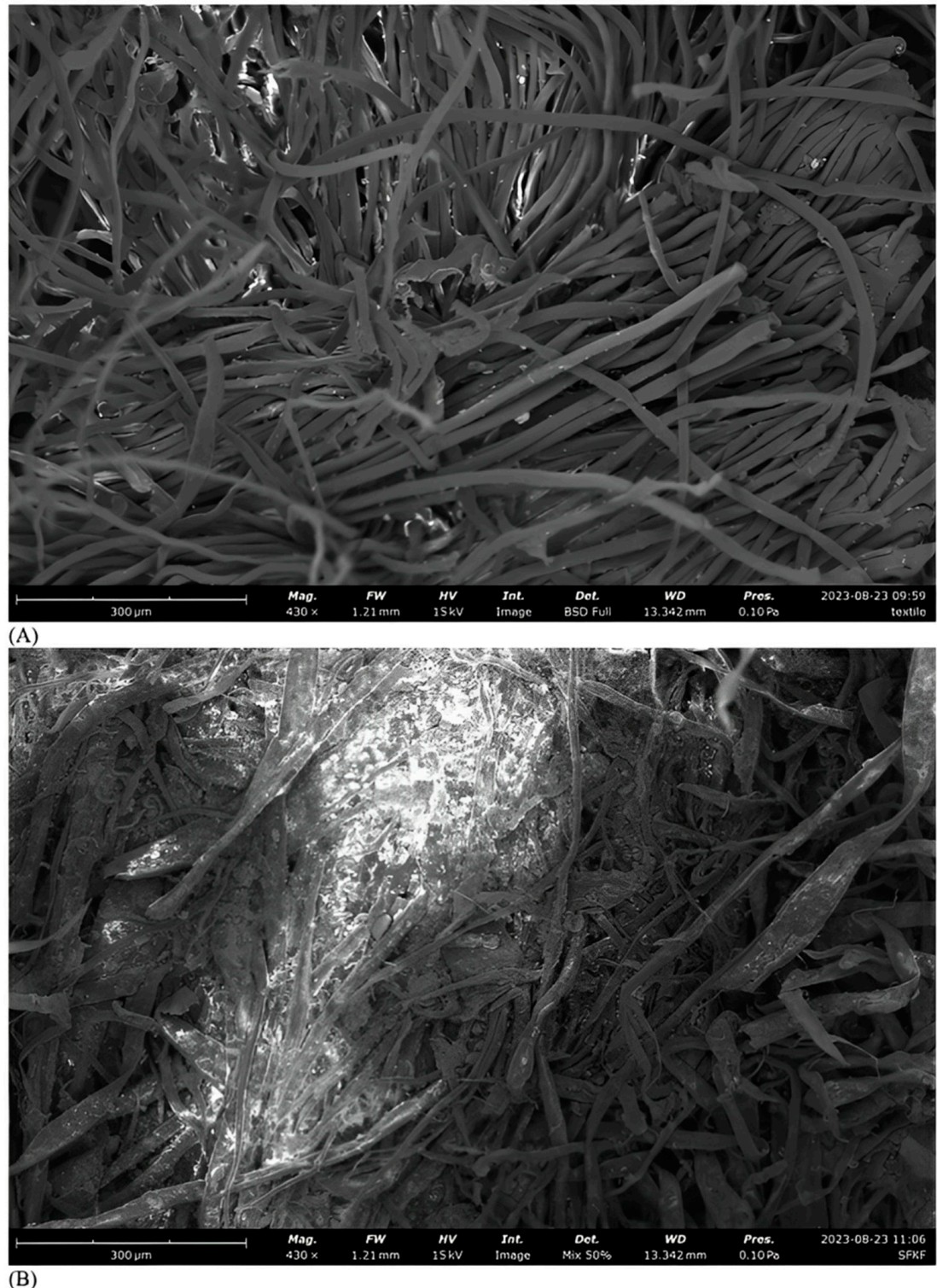

**Figure 3.** SEM images: (**A**) textile fibres and (**B**) kraft fibres.

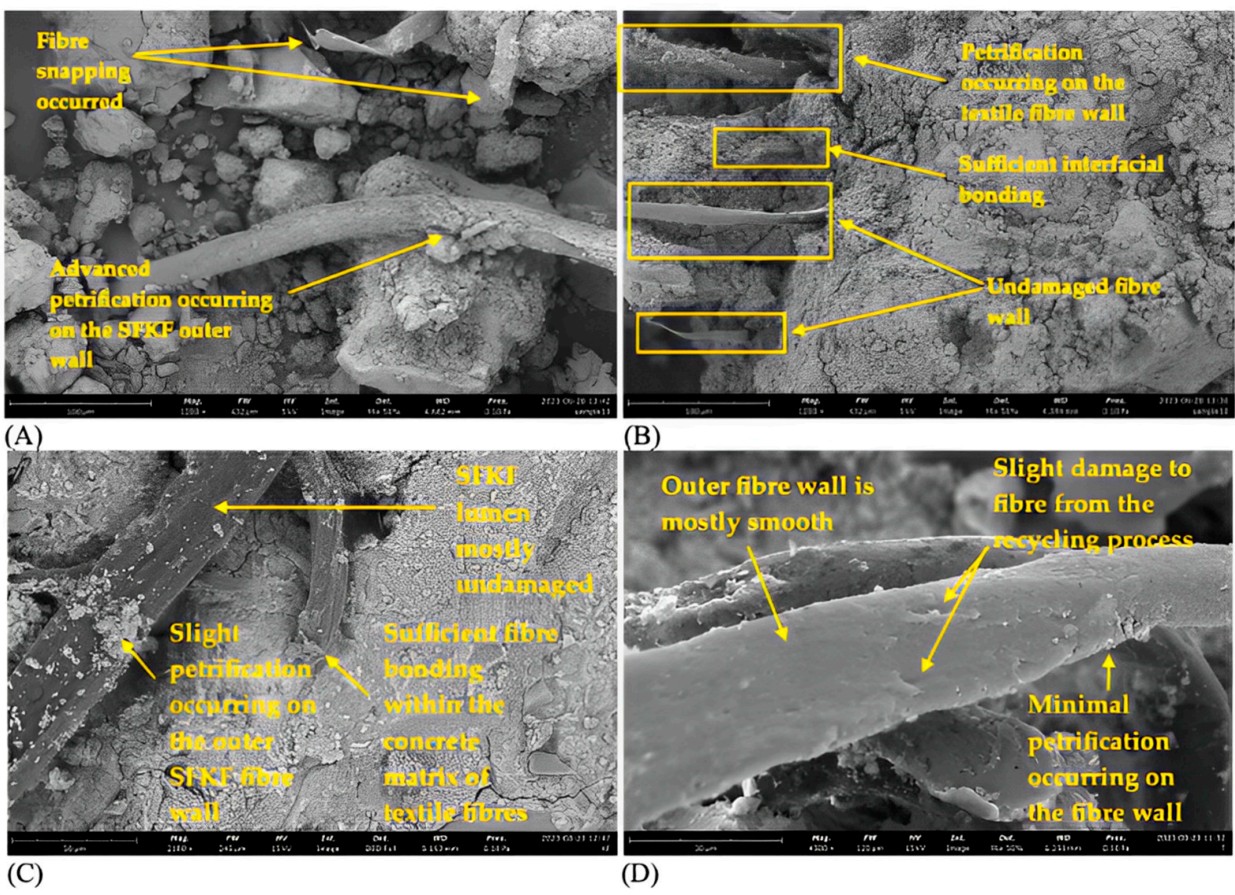

**Figure 4.** Concrete samples (**A**) S13, (**B**) S10, (**C**) S16, (**D**) S11.

Figure 4B highlights textile fibres with 5% SF in the concrete matrix. As shown, there was no damage to the outer fibre wall of the polyester fibres. This was primarily due to the chemical composition of polyester. Polyester is a polymer created from ester monomers, which are chemically inert [59]. This factor significantly reduces the reaction of the fibres within a highly alkaline environment, such as concrete. Polyester is hydrophobic, repelling water. This resistance to moisture prevents absorption and limits the potential for degradation of the fibres [60]. Figure 4D also illustrates textile fibres with 5% MK integration. Similarly to Figure 4B, there was minimal to no degradation to the outer fibre wall. Despite the recycling process of the polyester fibres, the fibres remained smooth, and their physical integrity was not hindered by the concrete materials. This is shown in how the shape remained mostly unharmed or distorted. In comparison, SFKFs are susceptible to damage to the outer fibre wall during the recycling process as well as when integrated in the concrete matrix. Despite pre-treatment of the fibrous material, the hydrophilic characteristics of SFKFs rendered the fibre susceptible to damage. This is shown in Figure 4C, where the fibres' shape and physicality are distorted compared to their synthetic counterparts. These imperfections in the SFKFs created enhanced anchorage points on the fibre walls. Their bonding capacity was increased due to the variation in shape and pattern of the microfibrils on the fibres' walls within the matrix [61]. The enhanced bonding of the fibres creates additional mechanical strength when under axial loading. Fibre composites mechanically fail under two conditions. Firstly, fibre pull-out occurs when the fibre does not have sufficient adhesion within the matrix. Secondly, fibre snapping occurs when the mechanical load exceeds the strength capacity of the fibre. The images suggest that fibre pull-out would occur with polyester fibres due to their physical characteristics and fibre snapping would occur with SFKFs.

The composition of S3 and S14 are shown in Figures 5 and 6, respectively. The SEM images with the EDS report demonstrate the weight concentration of the various chemical

elements within the concrete materials. The SEM image settings for Figures 5 and 6 have a magnitude (mag) of 500–700×, factor weight (FW) 1.04 mm, high voltage (HV) 15 kV, detection (Det) BSD full, within design (WD) 4.218 mm and pressure (Pres) 0.10 Pa. Figure 5 demonstrates the lower percentage of $Ca(OH)_2$ content due to the supplementation of gypsum and SFKFs within the matrix. This sample also highlights the silicon content from the SF pre-treatment of the SFKFs. Figure 6 illustrates the quantity of MK, SFKFs, and textiles in the composition of the concrete materials. As shown in the EDS report, there was a high content of aluminium and silicon. This demonstrates the sufficient agglomeration of all constituent materials in the matrix, highlighting adequate dispersion during the mixing process. MK contains a significant amount of aluminium, and its silicon content can be attributed to SF and fine aggregate materials.

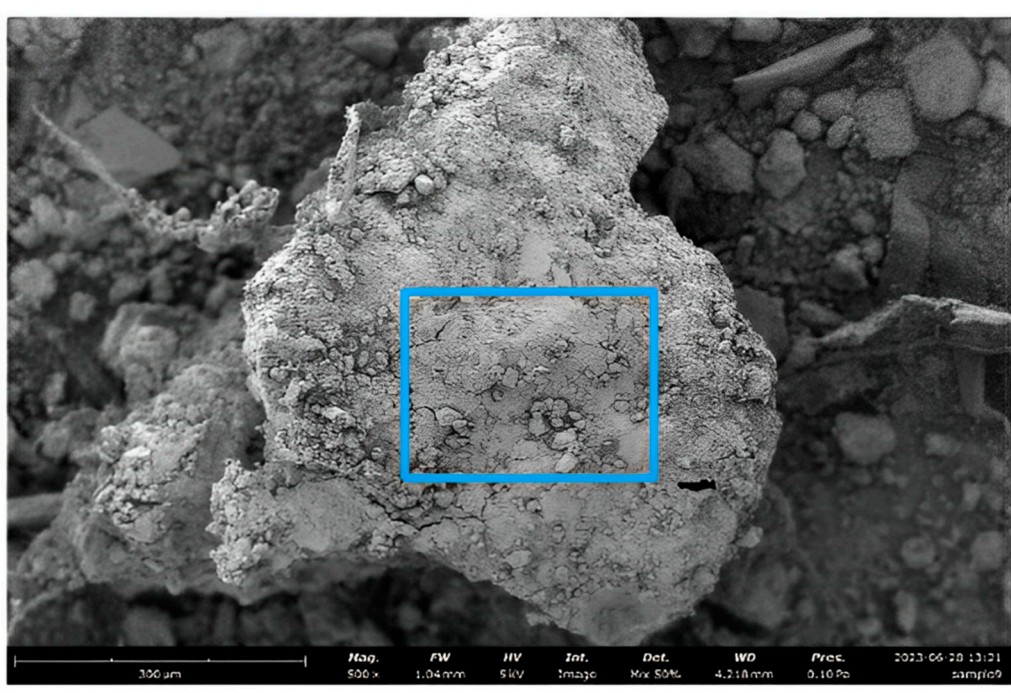

| Element | Atomic concentration | Weight concentration |
|---|---|---|
| Carbon | 0.0421907 | 0.013013 |
| Oxygen | 0.40683 | 0.167167 |
| Fluorine | 0.00615437 | 0.003003 |
| Silicon | 0.116558 | 0.0840841 |
| Calcium | 0.258684 | 0.266266 |
| Copper | 0.000613436 | 0.001001 |
| Zinc | 0.00178851 | 0.003003 |
| Niobium | 0.0738357 | 0.176176 |
| Palladium | 0.0340689 | 0.0930931 |
| Iodine | 0.0592758 | 0.193193 |

**Figure 5.** A targeted area of the material composition for S3.

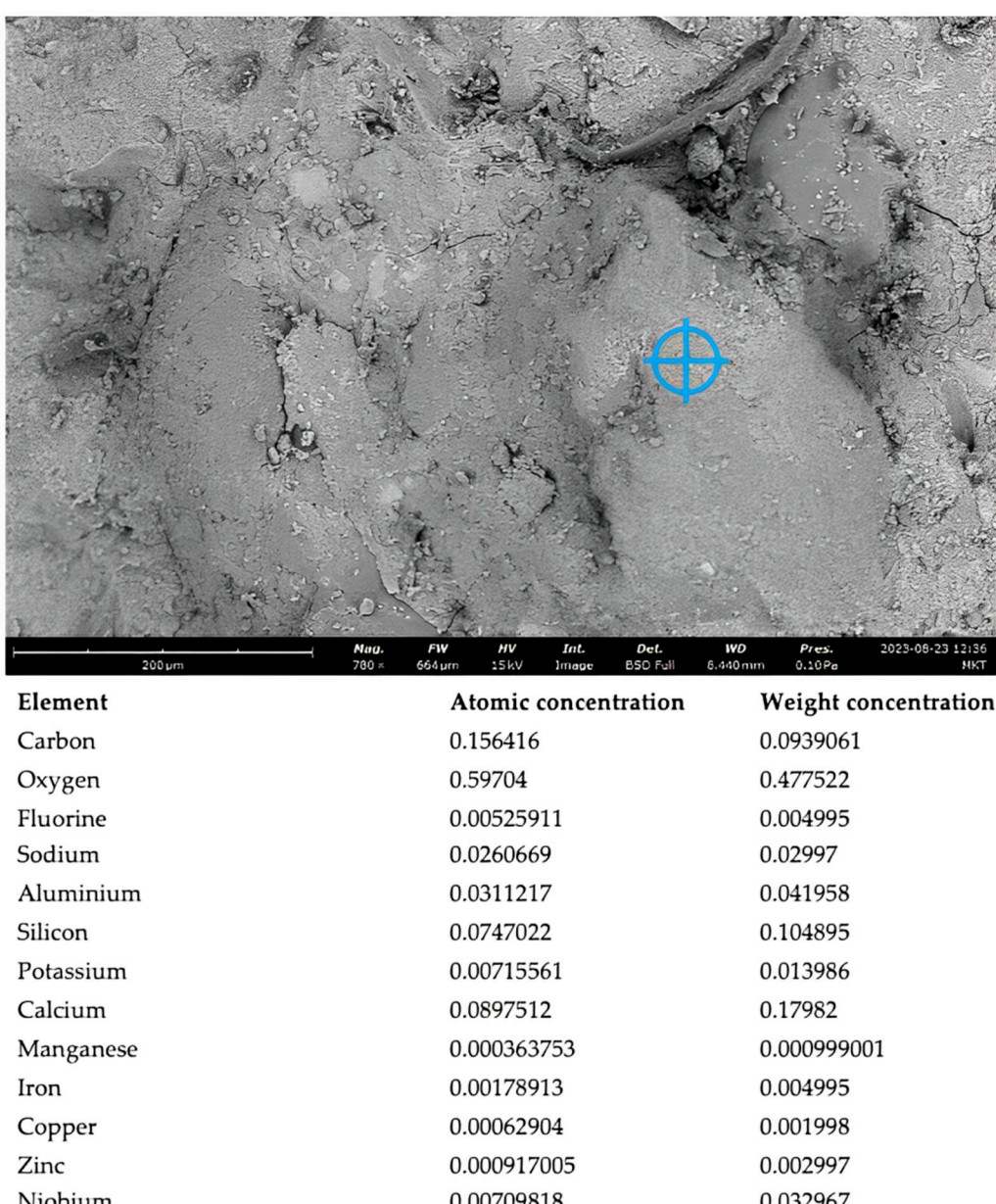

| Element | Atomic concentration | Weight concentration |
|---|---|---|
| Carbon | 0.156416 | 0.0939061 |
| Oxygen | 0.59704 | 0.477522 |
| Fluorine | 0.00525911 | 0.004995 |
| Sodium | 0.0260669 | 0.02997 |
| Aluminium | 0.0311217 | 0.041958 |
| Silicon | 0.0747022 | 0.104895 |
| Potassium | 0.00715561 | 0.013986 |
| Calcium | 0.0897512 | 0.17982 |
| Manganese | 0.000363753 | 0.000999001 |
| Iron | 0.00178913 | 0.004995 |
| Copper | 0.00062904 | 0.001998 |
| Zinc | 0.000917005 | 0.002997 |
| Niobium | 0.00709818 | 0.032967 |
| Palladium | 0.00169043 | 0.00899101 |

**Figure 6.** A targeted area of the material composition for S14.

### 5.2. Thermal Properties (TGA and DSC)

The TGA results are illustrated in Figure 7. The TGA experiments were conducted in a nitrogen atmosphere, employing a heating rate of 10 °C/min. Nitrogen was employed due to the inert atmosphere created. This assists in the prevention of oxidation and reduces the variables associated with the decomposition of the materials being studied. It is noteworthy that the overall thermographic profiles of the test samples remained consistent, regardless of the heating rate. These profiles exhibited the following distinct stages. First, initial water loss, primarily involving the desorption of physically bound water, occurred up to 100 °C. Secondly, the primary degradation phase, which included processes like dehydration reactions, extended until approximately 450 °C. Finally, there was a secondary degradation phase involving the carbonaceous residue, continuing until the conclusion of the experiment at 800 °C. The initial chart demonstrates S1, S2, S3, and S4. S3 degraded at a faster rate in comparison to the others due to the inclusion of 5% SFKFs in the mix design. This observation is in accordance with studies that identify three primary weight loss stages

with natural fibres [62,63]. Initially, a minor weight reduction occurred between 50 and 100 °C, attributed to the moisture evaporation within the fibre. Subsequently, hemicellulose and the primary constituent of lignin decomposed within the temperature range of 270–360 °C. Finally, the most substantial weight loss was observed as the cellulose component underwent significant degradation between 350–500 °C. It is interesting to note that S3 and S4 had a slower degradation at elevated temperatures due to the integration of gypsum compared to S1, the control. S2 contained 5% textile materials, but this did not adversely affect the weight loss at elevated temperatures. Due to the chemical composition of the textile materials being a polymer, the degradation was slower compared to natural fibres such as SFKFs. This is primarily associated with the inherent chemical stability of textile materials compared with natural fibres. The chemical bonds in synthetic polymers are more resistant to hydrolysis and oxidation compared to the complex organic molecules in natural fibres.

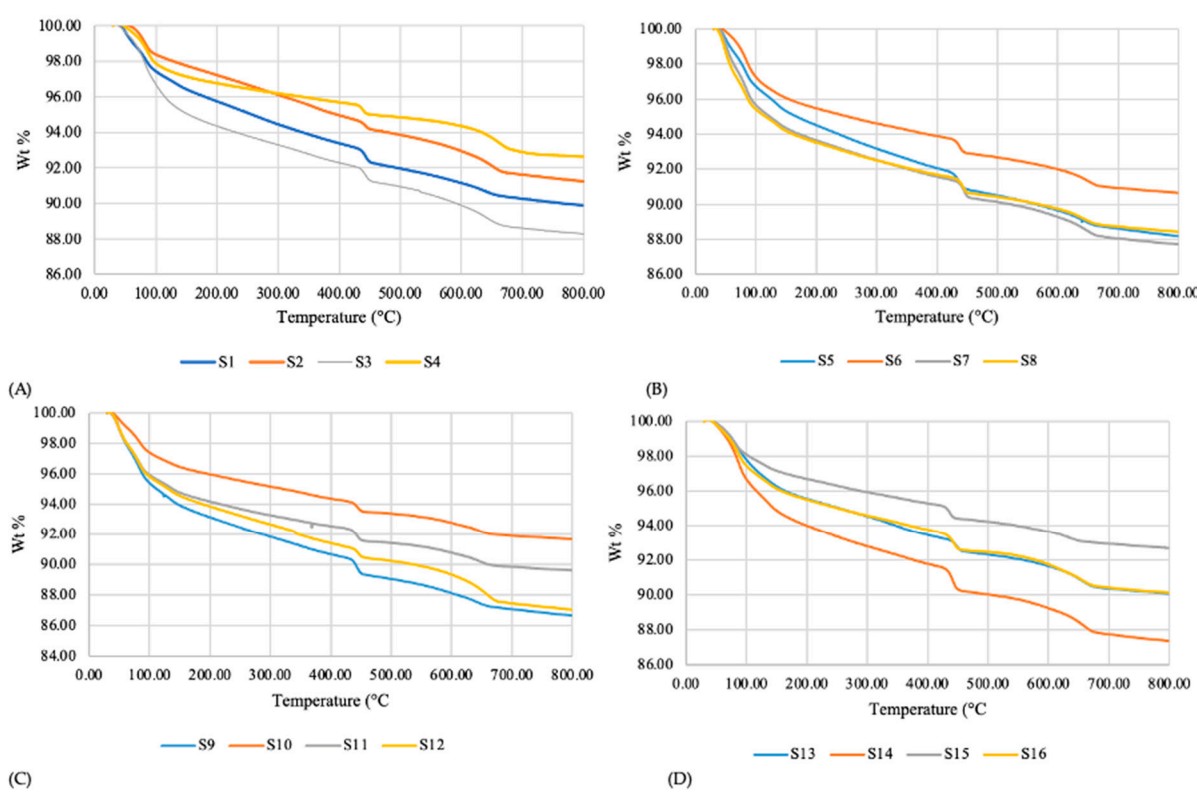

**Figure 7.** TGA results of samples 1–16 across four graphs (**A**–**D**).

S5, S7, and S8 had similar patterns of degradation due to the integration of MK and SF, with two distinct patterns of weight loss at 450 °C and 650 °C. However, the combination of gypsum and MK in S6 increased the durability of the composite at elevated temperatures in comparison. Despite the small percentage of gypsum (2.5%) mixed with 2.5% MK, gypsum can still reduce the degradation caused to the composite at higher temperatures. This was demonstrated with an increased weight of over 3% at similar temperatures compared to S5, S7, and S8. The least degraded fibre composite at elevated temperatures was S10, containing 5% textiles and 5% SF. Due to the chemical composition of textile materials and SF, S10 shows promising results for a more durable composite compared to its natural fibre counterparts. For example, 2.5% SFKFs in S9 resulted in a higher weight loss at the same temperatures as compared with S10, S11, and S12. In S10, S11 and S12, 5% textile fibres were included, with varied percentages of MK and SF. S12 contained both 5% MK and SF; however, increasing the cement replacement to 10% was shown to increase thermal degradation of the composite materials. This is due to the influence of the pozzolanic reaction, hydration kinetics, and microstructural development of the fibrous concrete

materials. Moreover, the reaction of Ca(OH)$_2$ to form C-S-H gels can be significantly affected by the integrated SCM percentage. The associated pozzolanic reactivity can affect the rate and extent of the thermal decomposition and weight loss at elevated temperatures. Additionally, the cement replacement can influence the porosity, pore distribution, and connectivity of pores. The can lead to changes in the transport of heat, gases, and moisture within the concrete during heating.

The integration of SF in the composite material demonstrates the enhanced durability of the composite material at elevated temperatures. This is illustrated with S15 and the integration of 5% SFKFs and 5% SF. Despite SF being used as a pre-treatment on the SFKFs, the thermal durability benefit was only increased when there was additional SF in the mix design. Moreover, SF increased the thermal durability at elevated temperatures when compared with MK. This was shown by S14 exhibiting prominent weight loss due to the addition of 5% MK when replacing cement. SF is an amorphous material composed of very fine, highly reactive particles, whereas MK is a pozzolanic material formed by the calcination of kaolin clay. SF is highly reactive due to its amorphous nature and smaller particle size, showing a more pronounced reactivity at lower temperatures than MK. MK in the samples reacts at higher temperatures due to the presence of the reactive alumina-silicate phases in the material. Moreover, SF exhibits a higher thermal stability due its amorphous structure and high melting point of >1600 °C. This is significantly higher than the calcination process of kaolin clay at 600–850 °C to form the material MK.

The TGA curve shows different endothermic or exothermic peaks, indicating various reactions taking place within each material. SF exhibited an exothermic reaction due to its high reactivity, whereas MK showed endothermic peaks during dehydroxylation and exothermic peaks during pozzolanic reactions at higher temperatures. The mass loss profile for both materials could differ significantly. SF showed a more gradual and continuous weight loss, while MK displayed distinct steps in weight loss corresponding to the dehydroxylation process and subsequent pozzolanic reactions. It is important to note that these behaviours can be affected by various factors like the specific characteristics of the materials, particle size, impurities, and the exact experimental conditions during TGA. The addition of both fibre materials (textiles and SFKFs) in S14, S15, and S16 caused varied thermal reactions due to the alternate percentages of SF and MK. For example, when MK was solely integrated, there was increased weight loss compared to the integration of SF, which demonstrates a reduction in weight loss caused by elevated temperatures. However, when both SF and MK were integrated at either 2.5% or 5% each for cement replacement, the degradation followed similar patterns. This is shown in S16 and S13. Despite the differences in the percentage of cement with SF or MK, the integration of these SCMs within concrete could produce similar degradation patterns due their similar pozzolanic activity, thermal stability, and microstructural changes. For example, SF could fill pores or voids in the concrete matrix, while MK contributed to the refinement of pore structure. Additionally, both materials could lead to the densification of the composite, which can create trends similar to the overall chemical microstructure of the samples. Both materials contain aluminium oxide and silicon dioxide, albeit at differing percentages. These results identify combinations of additive materials to utilize when integrated with fibre materials. Textile materials display a prominent thermal stabilization at elevated temperatures compared with SFKFs. However, the integration of SF can enhance the natural fibre durability and display similar patterns of weight loss as their synthetic counterpart.

The DSC results are illustrated in Figure 8. As expected, the DSC curves displayed two distinct peaks mirroring their corresponding TGA thermograms. The observed peaks were associated with endothermic processes, specifically in relation to two distinct phenomena. The first endothermic peak occurred between 100 °C and 130 °C. This absorption of heat corresponds to the release of physically bound water from the samples. The physically bound water evaporated or desorbed from the pore structure. Subsequently, higher temperatures indicated the release of chemically bound water. This first peak demonstrated the point of release of the physical presence of water within the composite materials. The sec-

ond endothermic peak, between 430 °C and 500 °C, was attributed to the primary pyrolysis reactions of the sample's main molecular chains. The downward valleys indicate exothermic reactions; however, there were only minor occurrences after the endothermic peak. The enthalpy of the samples was affected by the additive materials and their corresponding percentages. Moreover, the non-uniform signal in the enthalpy curves signified the variation of the mix design. The crystallinity of each constituent could be determined by the melting enthalpy of each material within the composition of the mix design, as shown in the second endothermic peak. Increasing the percentage of SF and MK was shown to reduce the heat flow as the temperature increased. S9 had 2.5% cement replacement with both SF and MK and demonstrated the highest endothermic peaks of all the samples. Increasing SF and MK to 5% cement replacement each stabilized the heat flow as the temperature increased. This was shown with S10, S11, and S12. Integrating fibrous materials demonstrates varied thermal reactions between the synthetic materials and their natural fibre counterpart. SFKFs in the concrete demonstrated a slower release in heat flow, as shown in S13, S15 and S16. However, the synthetic textile fibres had a larger thermal reaction corresponding to an increased endothermic peak, as shown with S14. This indicated the phase change of the materials. As the temperature increased, the phase pattern corresponded to the chemical reaction of the fibres. For example, the melting of the synthetic textile fibres led to crystallization of the materials within the matrix. Fibres with rapid crystallization rates may exhibit improved mechanical properties such as compressive strength, due to the formation of more densely packed crystalline regions. The DSC results demonstrated the melting point of the composite materials, which is exhibited via the endothermic peaks. The exothermic peaks are shown via the crystallization of the materials, then the amorphous transition takes place. When the textile fibres began to melt, the transition of fibre began with the crystallization of the fibres, then the amorphous transition took place. During the amorphous transition, the fibres became thermally stabilized.

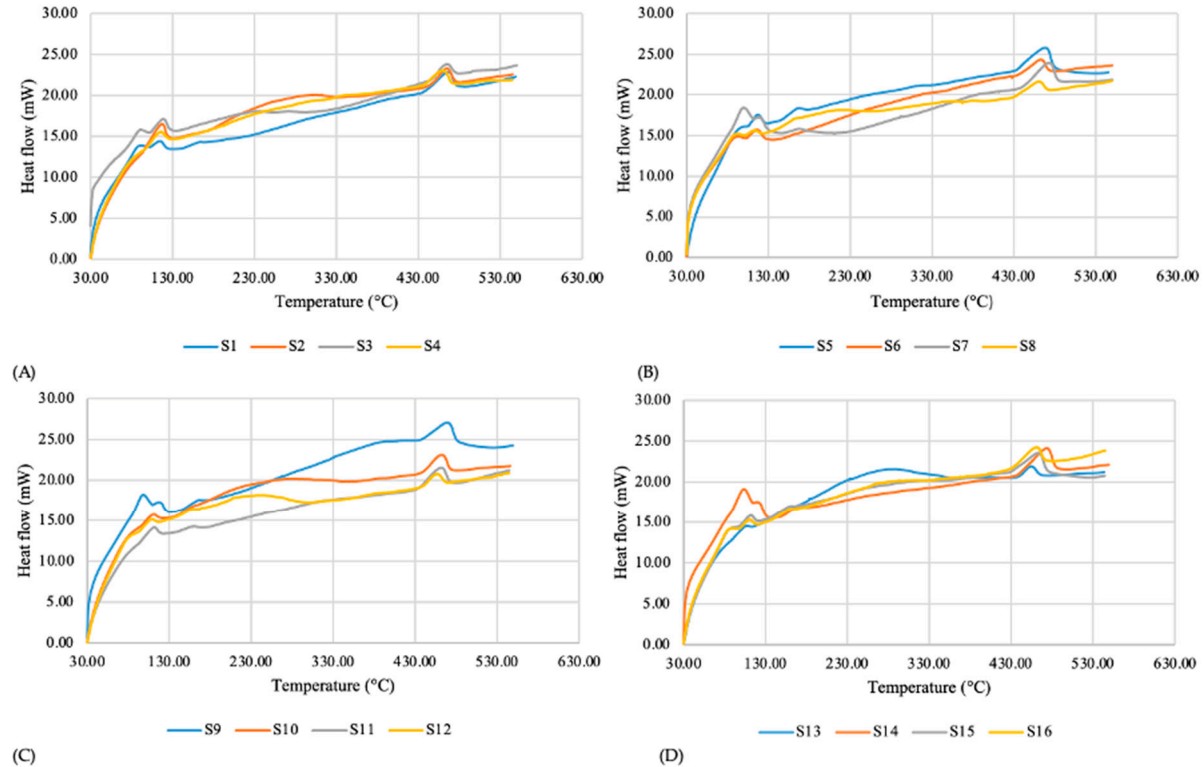

**Figure 8.** DSC results of samples 1–16 across four graphs (**A**–**D**).

## 6. Conclusions and Future Research

The building and construction industry relies heavily on concrete and cementitious composites due to their exceptional attributes, including strength, durability, and versatility. However, the widespread use of these materials has significant environmental drawbacks, such as resource depletion, carbon emissions, and waste accumulation. The current study highlights the need to explore alternative composite designs when using waste fibrous materials in composite designs, specifically focusing on alternative additives used to sustain and enhance the durability of fibrous materials. One of the key areas of this study is the thermal investigation of multiple fibre types in various concrete matrices. The research aims to understand thermal stability when using both textile and cardboard fibres. While previous studies have explored single fibre types in concrete, there is limited research on the combination of multiple fibres in different concrete matrices. This study demonstrates that materials such as gypsum, metakaolin, and silica fume, when integrated into concrete, can enhance the durability of the composite at elevated temperatures. These materials reduce the production of $Ca(OH)_2$ in the concrete matrix, which can degrade fibrous materials. This leads to an increase in the longevity of the fibres and can contribute to the mechanical strength of the composite. The study also highlights that the integration of fibres, both synthetic and natural, has varied thermal reactions, and the choice of additives plays a crucial role in the overall performance of the composite. These findings contribute to the development of more sustainable and durable construction materials, helping to address the environmental challenges associated with the construction industry. Future researchers can utilize these findings to determine additive and fibre percentages within mix designs, reducing extensive sample testing. Future research can be directed toward complex fire testing methods, mechanical and durability experiments. The most important outcomes of this research are summarized as follows:

- Reduced thermal degradation when using gypsum in concrete materials.
- Using both SF and MK have a stable thermal reaction despite varied percentages.
- Natural fibres degrade significantly faster than textile fibres at similar temperatures.
- SF enhances composite thermal durability more than MK.
- SFKFs have a slower heat flow in response to the increased temperature.
- Textile fibres have higher endothermic peaks corresponding with crystallization.

**Author Contributions:** Conceptualization, R.H.; methodology, R.H.; software, R.H. and M.A.; validation, R.H., P.J. and M.A.; formal analysis, R.H.; investigation, R.H.; resources, R.H.; data curation, R.H.; writing—original draft preparation, R.H.; writing—review and editing, R.H. and M.S.; visualization, R.H.; project administration, M.S.; funding acquisition, M.S., P.J., E.Y., S.S. and Z.V. All authors have read and agreed to the published version of the manuscript.

**Funding:** This research was funded by Sustainability Victoria, Victoria for funding this project under the grant scheme Circular Economy Markets Fund: Materials; grant number C-12713.

**Institutional Review Board Statement:** Not applicable.

**Informed Consent Statement:** Not applicable.

**Data Availability Statement:** The data presented in this study are available on request from the corresponding author.

**Conflicts of Interest:** The authors declare no conflicts of interest.

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
