# Peer review of "The Thermo-Phase Change Reactivity of Textile and Cardboard Fibres in Varied Concrete Composites"

_sustainability, doi:10.3390/su16083221_

Round 1

Reviewer 1 Report

Comments and Suggestions for Authors

This paper describes a study that explores the concrete's thermal reactivity with the additive of textile and cardboard fibers. This work is beneficial to the community for further understanding of the thermal stability of various fiber materials in the concrete materials. Therefore, I recommend its acceptance after some minor revision.

1. The bottom texts in the SEM images are too blurred to read. 

2. The same issue is in the EDS report. The text and number are hard to read. 

3. It seems to me that the authors used the characterization instrument generated images directly and insert those images to this paper, which is fine as long as the reader can read them clearly. However, in this case, I find it difficult to read the exact texts and numbers in the images, either SEM, EDS or DSC. I suggest the authors plot the graph themselves instead of using raw images from the measurement software.

Author Response

Thank you for your valuable comments. 

The attached provide a detailed response to the reviewer's comments.

We sincerely hope that the reivsion meets the stnadard for publication. 

Thank you 

Reviewer 2 Report

Comments and Suggestions for Authors

The manuscript offers a thorough examination of the prospective incorporation of textile and cardboard fibers as supplements in concrete composites, with a specific emphasis on their thermo-phase change characteristics. This subject matter holds relevance, addressing significant environmental considerations within the construction sector. The following suggestions are provided to improve the manuscript's clarity.

Please provide more details on the SEM settings and magnification used for obtaining the images in Figure 4.

How does the pre-treatment of SF contribute to the longevity of SFKFs in the concrete matrix?

In Image (C), what is the significance of the distorted shape and physicality of SFKFs compared to their synthetic counterpart?

Why was a nitrogen atmosphere chosen for TGA experiments, and how does it impact the observed thermographic profiles?

How does the chemical composition of textile materials contribute to the slower degradation compared to natural fibers like SFKFs, as discussed in the TGA results?

Can you explain the observed variations in weight loss at elevated temperatures between S9 and S10, considering both contain 5% textile fibers but with different cement replacements?

What specific characteristics of SF and MK contribute to their different thermal reactions, as highlighted in S14 and S15?

In S16 and S13, where both SF and MK are integrated, why do the degradation patterns follow similar trends despite different percentages of cement replacement?

What is the significance of the endothermic peaks observed between 100°C and 130°C in the DSC curves, and how do they relate to the physically bound water in the samples?

Provide more insights into the observed phase change indicated by the DSC curves, particularly in relation to the melting and crystallization of synthetic textile fibers.

To enhance the introduction, it would be beneficial to expand upon existing research to better contextualize the study within the broader landscape of concrete composite materials. Integrating the following studies into the introduction will offer a more comprehensive overview, underscoring the significance of investigating the utilization of waste materials in the development of sustainable construction materials. This approach will not only emphasize the relevance of the study within current research trends but also elucidate the specific research gap that the manuscript aims to address.

-Corrosion effect of acid/alkali on cementitious red mud-fly ash materials containing heavy metal residues

-Establishing a 3D aggregates database from X-ray CT scans of bulk concrete

-Mixed-Mode Debonding in CFRP-to-Steel Fiber-Reinforced Concrete Joints

-Exploring temperature-resilient recycled aggregate concrete with waste rubber: An experimental and multi-objective optimization analysis

Author Response

(The authors gave the same response as above.)

Reviewer 3 Report

Comments and Suggestions for Authors

Dear authors,

The object of study presented in this manuscript is of scientific and technological interest and could be a good contribution to the state of the art of sustainable concrete and indeed to the journal. However, to consider the publication of this manuscript, it must be improved both from the point of view of format and scientific rigor and to resolve certain aspects that I consider serious. Below, I provide various comments and suggestions that I hope will help improve the manuscript. If these aspects are resolved, its publication can be evaluated.

After reviewing the manuscript, I have some comments, doubts, and suggestions:

Introduction section: The introduction needs improvement, especially in terms of text structure. Text from line 53 to 60 can be avoided. To follow the argument of the text, it should start with the state of the art of the topic under study, including its drawbacks, advances, etc., followed by the objective of the work, relevance, and contribution to the field. In general terms, the manuscript follows this structure, but it needs to be clarified further.

General aspects:

All figure and table captions must be revised to be as self-explanatory as possible. Include more detailed information, such as units, scales, or whatever is appropriate in each case.

Correct the format of the numbered sections; from section 3 onwards, the manuscript uses 3.0, etc.

Methodology section:

In Table 1, please define the abbreviations in the caption, and specify whether percentage units are in wt.% or vol.%.

Table 2 is not mentioned in the text. How was this data obtained? Please define this.

The origin of the samples must be specified, and the chemical composition of all of them should be added.

How was the amount of sample and the number of samples for each type prepared?

TGA: Define the model of Mettler-Toledo. Is it a simultaneous TGA-DSC? Define the amount of sample analysed, type of crucible, number of replicas, and measurement accuracy. Specify whether the heating rate is 10K/min or 60K/min.

DSC: Similar comments as for TGA. Additionally, specify the gas flow.

Results section:

Replace "Image (B)" in line 256 with "Figure 4 b)" throughout the text. Revise the entire manuscript accordingly.

The discussion and conclusions extracted from the SEM images need to be corroborated by complementary techniques.

Figure 4 needs improvement. Add SEM scales and reconsider the appearance of the yellow text.

Revise the format of figures 5 and 6 and the accompanying text; there seems to be a formatting error. Please review this section thoroughly.

The title of subsection 5.2 "Thermal and calorimetric properties" is redundant, as DSC and TGA measures thermal properties.

TGA results: Define the TGA profiles obtained and depicted in figure 7, which represent weight loss as a function of temperature. Normalize Figure 7 to 100 wt%. Add corresponding letters (a), (b), (c), (d) to each image.

DSC results: In line 378, change "The differential calorimetry of the samples..." to "The enthalpy of the samples...". Explain the non-uniform signal in the enthalpy curves and consider determining the enthalpy of the peaks for proper comparison of results. The DSC analysis of results becomes incomplete.

The figures need improvement like Figure 7.

Regarding the most relevant results, how many repetitions and samples were used for each study? If only one sample per type was analysed, it is insufficient for publication, especially considering the possibility of drawing correct conclusions, particularly in this study where non-homogeneous mixtures are involved.

Finally, in my opinion, this study would benefit from including mechanical strength results to enhance its completeness and justify certain statements made throughout the manuscript.

Author Response

(The authors gave the same response as above.)

Reviewer 4 Report

Comments and Suggestions for Authors

This study aims to understand the thermal stability when using both textile and cardboard fibers in varied concrete composites. The topic is interesting. The outcomes and conclusions are for researchers in concrete science. I have following observation for the improvement of manuscript 

Improvement is needed in the introduction section to better highlight the novelty and contribution of the study. Additionally, there are some inconsistencies in lines 80-82, which create confusion for readers. For example, It is important to differentiate between additives and alternative additives as they refer to two distinct concepts. Furthermore, it is advisable to include the structure of the manuscript at the end of the introduction to provide readers with a clear overview of the organization and flow of the paper. Lastly, it is recommended to remove the duplicate figures, specifically Figure 5 and Figure 6, as their redundancy may cause confusion for reader.

Author Response

(The authors gave the same response as above.)

Round 2

Reviewer 2 Report

Comments and Suggestions for Authors

The revised manuscript is acceptable.